# Profile Resemblance in Health-Related Markers: The Portuguese Sibling Study on Growth, Fitness, Lifestyle, and Health

**DOI:** 10.3390/ijerph15122799

**Published:** 2018-12-10

**Authors:** Sara Pereira, Peter T. Katzmarzyk, Donald Hedeker, José Maia

**Affiliations:** 1CIFI2D, Faculty of Sport, University of Porto, 4200-450 Porto, Portugal; sara.s.p@hotmail.com; 2Pennington Biomedical Research Center, Louisiana State University, Baton Rouge, LA 70808, USA; peter.katzmarzyk@pbrc.edu; 3Department of Public Health Sciences, University of Chicago, Chicago, IL 60637, USA; DHedeker@health.bsd.uchicago.edu

**Keywords:** siblings, health behaviors, physical fitness, body composition, adolescents

## Abstract

The co-occurrence of health-related markers and their associations with individual, family, and environmental characteristics have not yet been widely explored in siblings. We aimed to identify multivariate profiles of health-related markers, analyze their associations with biological, sociodemographic, and built environment characteristics, and estimate sibling resemblance in these profiles. The sample includes 736 biological siblings aged 9–20 years. Body fat was measured with a portable bioelectrical impedance scale; biological maturation was assessed with the maturity offset; handgrip strength, standing long jump, one-mile run, and shuttle run were used to mark physical fitness. Health behaviors, sociodemographic, and built environmental characteristics were recorded by questionnaire. Latent profile analysis and multilevel logistic regression models were used; sibling resemblance was estimated with the intraclass correlation (*ρ*). Two multivariate profiles emerged: “P1 = fit, lower fat and poorer diet” (86.7%) and “P2 = higher fat and lower fit, but better diet” (13.3%). Siblings whose fathers were less qualified in their occupation were more likely to belong to P2 (OR = 1.24, *p* = 0.04); those whose fathers with Grade 12 and university level education were more likely to fit in P2 compared to peers living with fathers having an educational level below Grade 12 (OR = 3.18, *p* = 0.03, and OR = 6.40, *p* = 0.02, Grade 12 and university level, respectively). A moderate sibling profile resemblance was found (0.46 ≤ *ρ* ≤ 0.55). In conclusion, youth health-related markers present substantial differences linked with their body composition, physical fitness and unhealthy diet. Furthermore, only father socio-demographic characteristics were associated with profile membership. Sibling´s profile resemblance mirrors the effects of genetics and shared characteristics.

## 1. Introduction

Behaviors like physical inactivity, sedentariness, and unhealthy dietary intake are risk factors associated with increases in the worldwide incidence of non-communicable diseases [1]. On the contrary, the adoption of health-related behaviors like regular involvement in moderate-to-vigorous physical activities, consuming healthy diets, combined with adequate physical fitness [2], and healthy body fat levels [3] are likely to reduce the risk of premature death by non-communicable diseases. These markers, both healthy and unhealthy, are acquired and developed during childhood within the family orbit, become more pronounced during adolescence, and tend to persist into adulthood [4].

### 1.1. Area of Residence

There is growing evidence suggesting that geographic area of residence is an important correlate of healthy youth development, since it may provide opportunities, as well as conditions that impact health [5]. Further, a domain of current interest is the built environment and how access to public transportation, shops/markets, traffic, safety, and available areas for recreation or physical activity practice can contribute to individual health [6,7,8]. For example, Mitas et al. [6] showed that adolescents who perceived their neighborhood environment as the safest were significantly more likely to meet the recommendations for leisure-time walking. Furthermore, safe neighborhoods encourage and facilitate youth involvement in unsupervised sports´ participation [6]. Sallis et al. [7] reported that walkability was positively and significantly related to objectively-measured physical activity and negatively related to sedentary time and TV time in adolescents aged 12–16 years. 

### 1.2. Familial Environment

The family environment also plays a role in the acquisition and development of lifestyle behaviors in youth, with significant impacts on their health [9,10]. Information gathered on nuclear families, twins, or siblings is a useful way to better understand the magnitude of environmental and genetic factors on lifestyle behaviors and other health-related markers. For example, Nelson et al. [11] used sibling data to show that household environments accounted for 8%–10% of the total variation in adolescent fast food intake and sedentary behaviors and 50% of the total variation in adolescents being overweight. Additionally, a systematic review by Fisher et al. [12] of twin studies reported that the shared environment had a strong effect size on physical activity levels (60%), with a smaller contribution from genetic factors (21%). 

### 1.3. Clustering

Much of the available research has not considered predictors from both the neighborhood and the family environment, or even in association with correlates from other domains, of the expression of behaviors and traits in youth. Latent profile/class analysis is a very useful way to tackle this issue, given that it allows for the classification of individuals into distinct profiles/classes based on the co-occurrence of traits and behaviors [13]. For example, based on physical activity and sedentary variables, Patnode et al. [14] identified three distinct classes for boys and girls with significant differences between classes for a number of demographic indicators. Furthermore, by adding dietary intake information, Iannotti and Wang [15] identified three distinct classes whose membership was related to age, sex, race/ethnicity, and socioeconomic status. Similarly, Pereira et al. [16] added data on sleep time and were only able to identify two distinct classes significantly related to individual characteristics (sex and maturity offset), but not to sociodemographic factors (maternal education and household income). 

### 1.4. Profile and Sibling Approach

We were not able to identify studies that simultaneously focused on lifestyle behaviors and other traits related to health (e.g., physical fitness and body composition) and that also explored the roles of biological, socio-demographic characteristics, and built environment in the prediction of profiles/class membership. Further, given population heterogeneity regarding health outcomes, which are governed by a plethora of factors (biological, behavioral, environmental, genetic, and cultural), we believe that using a person-centered approach will provide strong clues to identify putative profiles for different people, which, in turn, may provide novel insights and a holistic view regarding health-related markers and ways to develop and promote them.

Siblings share a substantial part of their life, but may also exhibit different developmental trajectories and healthy/unhealthy lifestyle profiles. Thus, using siblings as a template to explore the co-occurrence of lifestyle behaviors and other traits related to health may provide important clues about the effects of common and unique characteristics that are associated with youth health. Consequently, relevant information may be provided to families, educators, and health professionals in terms of prevention and planning interventions. Hence, the present study aims: (1) to identify multivariable profiles of health-related markers, namely lifestyle behaviors (physical activity, healthy, and unhealthy food habits), physical fitness, and body composition; (2) to investigate the associations among biological, sociodemographic, and built environment characteristics and the multivariable profiles; (3) to estimate sibling similarity in these multivariable profiles. Based on these, the following questions were addressed: (1) Can we identify multivariate profiles of health-related markers in youth? If so, how many will emerge? (2) How are biological, sociodemographic and built environment characteristics associated with profile membership, i.e., how strong are their effect sizes? (3) How sizeable is sibling resemblance in their multivariate profiles?

## 2. Materials and Methods

### 2.1. Study Participants

The current study sample was from the Portuguese Sibling Study on Growth, Fitness, Lifestyle and Health, which investigates physical growth, body composition, physical fitness, physical activity, metabolic syndrome, and health behaviors in a cohort of siblings [17]. The subjects in this study were part of a larger study, the Portuguese Healthy Family Study [17], in which participants were randomly sampled from schools in mainland (north and central) Portugal and from the Azores islands. Those who had their siblings studying in the same school were invited to take part in the study (~4100), and the response rate was ~80%. Written informed consent was obtained from legal guardians, and the project was approved by the Ethics Committee of the University of Porto and school authorities (CEFADE 09.2015). The final sample consisted of 3285 siblings 9–20 years of age. However, for the present study, the sample comprised 736 biological siblings (350 females and 386 males) from 370 nuclear families with complete data on lifestyle behaviors, physical fitness, and body composition. No statistically-significant mean differences were observed between included and excluded siblings in height, weight, and % body fat. 

### 2.2. Health-Related Markers

#### 2.2.1. Body Composition

Percentage body fat (%Fat) was estimated with youth in light clothing and using a reliable and valid instrument [18], a portable bioelectrical impedance scale (TANITA BC-418 MA Segmental Body Composition Analyzer Tanita Corporation, Tokyo, Japan).

#### 2.2.2. Physical Fitness

Physical fitness was assessed with four tests: (1) handgrip strength, using a hand dynamometer (Takei Digital Grip Strength Dynamometer, Model T.K.K.5401, Tokyo, Japan) with participants gripping the dynamometer with maximum force for 5–10 s; (2) standing long jump, where participants jumped as far as possible from a standing position; (3) 1-mile run/walk test, where participants ran/walked the distance in the shortest time possible; (4) shuttle-run (SHR), with participants running as fast as possible to another line (9 m away) where two small blocks were placed, picking up a block, returning to place it behind the starting line, and then repeating the route for the second block. A standardized *z*-score for each physical fitness test result was computed, which were summed to obtain a physical fitness *z*-score for each subject. The signs in the 1-mile run/walk and shuttle-run were reverted. These tests have been shown to be reliable and valid [19]. 

#### 2.2.3. Dietary Intake

Diet intake data were obtained from a food frequency questionnaire (FFQ) adapted and modified from the Health Behavior in School-aged Children Survey (HBSC) [20] using typical Portuguese food items. This questionnaire was previously used in multi-country studies [21]. Youth were asked about various types of food consumed in a typical week. For each item, the reported answers were converted into weekly portions as follows: “never” = 0; “less than once per week” = 0.5; “once per week” = 1; “2–4 days per week” = 3; “5–6 days per week” = 5,5; “once a day, every day” = 7; and “more than once a day” = 10, as previously advocated [22]. Food items related to healthy diet were as follows: fruits, vegetables, dark-green vegetables, orange vegetables, fruit juice, skim milk, low-fat milk, whole milk, cheese, other milk products, bread or whole grains, beans, lentils, bean curd, eggs, fish. Food items related to unhealthy diet were as follows: sweets, sugar-sweetened sodas, cakes, pastries, donuts, diet sodas, ice cream, potato chips, French fries, fast foods, sports drinks, energy drinks, fried food. Portions of each food item were summed, and two scores were derived for healthy and unhealthy diets. 

#### 2.2.4. Physical Activity

Total physical activity (TPA) was obtained with the Baecke questionnaire [23], a reliable and valid instrument [24]. This questionnaire includes three specific domains based on a total of 16 questions: work/school PA, leisure-time PA, and sports participation. TPA was estimated based on the sum of these three specific domains. For each domain, each score ranged from 1 (minimal) to 5 (maximal), such that the TPA score varied between 3 and 15. Participants answered the questionnaire during regular physical education classes under the supervision of their school-teacher, as well as by a trained research team member.

#### 2.2.5. Screen Time

Information about screen time was obtained using the U.S. Youth Risk Behavior Surveillance Survey [25] questionnaire by self-administered questions: “How long do you watch TV per day?” and “How long do you use your computer or play non-active video games per day?” Answers ranged from <30 m, 30 m–1 h, 1 h–1 h 30, 1 h 30–2 h to >2 h, subsequently categorized from 0–4 (−/+). Individual scores were summed and obtained a total score for screen time, as reported in different studies [26,27,28]. 

### 2.3. Biological Maturation

Biological maturation was assessed with the maturity offset [29], which estimates the temporal distance (in decimal years) from age-at-peak height velocity (PHV). A positive (+) maturity offset indicates the number of years the participant is beyond PHV, whereas a negative (−) maturity offset represents the number of years the participant is before PHV. This method has been widely used in children and youth [16,30,31].

### 2.4. Sociodemographic Characteristics

Sociodemographic characteristics were based on the occupation and education of both parents (mothers and fathers). Parents’ occupation was categorized into ten groups (from 0–9) according to the Portuguese National Classification of Occupations (2010), where Group 0 is the highest socioeconomic status (SES) and Group 9 is the lowest. Categories are as follows: (0) armed forces; (1) central administration/politicians and executive directors; (2) specialists of intellectual and scientific activities; (3) technicians and intermediate-level jobs; (4) back-office jobs; (5) security, seller, and individual services; (6) farmer and qualified workers of farm, fish, and forest; (7) qualified industry and building jobs: (8) machine and equipment operators; and (9) nonqualified jobs. Parents’ education was obtained according to the following categories: (1) <Grade 12; (2) Grade 12/diploma for technical qualification (equivalent to high school); (3) university level. 

### 2.5. Built Environment

Built environment information was obtained via questionnaire. We applied the Portuguese version of the Environmental module (environmental perception of the residential area) of the International Physical Activity Study, a reliable and valid instrument [32,33], previously used in the Portuguese population [34]. This questionnaire includes questions about the traffic system, accessibility to public transportation and shops/markets, housing density, perceived safety of the neighborhood, the presence of sidewalks and bike paths, and recreational facilities. The response item options were as follows: completely disagree, partially disagree, partially agree, or completely agree. For the purposes of this study, options were dichotomized into two categories: 0 = disagree (completely disagree, partially disagree); 1 = agree (partially agree or completely agree).

## 3. Statistical Analysis

Descriptive statistics were computed as means (±standard deviations) and frequencies. Then, as advocated [35,36], we removed the effects of age and sex from the six health-related markers (percent body fat, global physical fitness, physical activity, healthy and unhealthy food habits) using a multiple regression analysis. Residuals from these regressions were standardized using a *z*-score transformation. These analyses were done in SPSS 23. Then, these standardized residuals were exported to Mplus software 7.4 in order to identify the best fitting number of latent profiles using iterative maximum likelihood estimation techniques [37]. A series of models was fitted to the data with a sequence of putative profiles, and as advocated, we relied on the Akaike information criteria (AIC), the Bayesian information criteria (BIC), as well as the Vuong–Lo–Mendell–Rubin likelihood ratio test to compare models and determine the number of latent profiles [38]. Further, we also relied on recommendations from Geiser [39] to set the algorithm iterations to avoid local maxima.

Using the best solution on profiles and given the nested structure of the data (subjects nested within sib-ships), we used a multilevel logistic model to predict profile membership (P1 is the reference) and sequentially-tested three models of increasing complexity. Model 1 (M_1_) only included biological maturation; in Model 2 (M_2_), we added father and mother education, as well as their occupation; finally, Model 3 (M_3_) included built environment covariates. Parameter estimates were obtained via maximum likelihood as implemented in Stata 13 software. Model comparison was done with the deviance statistic as is customary [40]. Differences in deviances follow a Chi-squared distribution with degrees of freedom equal to differences in the number of parameter estimates. Finally, intraclass correlations (*ρ*) and the corresponding 95% confidence intervals (95% CI) as suitable measures of sibling similarity were computed from the variance components [40].

## 4. Results

Descriptive statistics for biological characteristics, body composition, physical fitness, lifestyle behaviors, sociodemographic characteristics, and built environment are presented in Table 1. On average, sibling pairs had similar chronological ages and biological maturation. However, sister-sister (SS) pairs had more body fat than brother-brother (BB) and brother-sister (BS) pairs. Further, BB pairs were more fit and active, but had higher screen time and higher unhealthy diet scores than SS and BS pairs. Healthy diet scores were similar across all types of siblings. A slight difference was observed for mothers’ and fathers’ occupation, favoring BS pairs. For education level, the highest percentages of responses for BB and BS pairs were observed in “Grade 12/diploma/technical qualification” for both the mother and father, while for SS pairs, the highest percentages category was “<Grade 12”, also in mother and father. Regarding the built environment, the perception about all items was similar in all sib-types. 

Figure 1 illustrates the best profile solution merging lifestyle behaviors (physical activity, screen time, and healthy/unhealthy food) and biological markers (physical fitness and body fat). Based on the statistical tests previously mentioned (AIC, BIC, and the Vuong–Lo–Mendel–Rubin likelihood ratio test) (Table 2), a two-latent profile model best fitted the data, and the counts and percentages of individuals in each profile were: Profile 1, *n* = 638 (86.7%), and Profile 2, *n* = 98 (13.3%). We labeled Profile 1 (our reference profile) as “fit, lower fat, and poorer diet” and Profile 2 as “higher fat and lower fit, but better diet”. 

Table 3 displays results from the multilevel logistic regression models. In M_1_, results showed that more mature siblings were less likely to belong to Profile 1 (OR = 1.18 ± 0.10, *p* < 0.05). In M_2_, with family-level covariates, a better fit was obtained relative to M_1_ (χ^2^ = 84.26 with 6 df, *p* < 0.001). In this model, biological maturation ceased to be a significant predictor of profile membership (*p* > 0.05). Siblings whose fathers were less qualified in their occupation were more likely to belong to Profile 2 (OR = 1.24 ± 0.12 *p* < 0.05). Additionally, those whose fathers had Grade 12 and university-level education were also more likely to fit in Profile 2 compared with peers living with fathers having below Grade 12 education (OR = 3.18 ± 1.72 and OR = 6.40 ± 5.09, *p* < 0.05, Grade 12 and university level, respectively). Built environmental covariates were added in Model 3, but this model did not improve the model fit over M_2_ (χ^2^ = 4.28 with 6 df, *p =* 0.135). This indicates that built environmental covariates did not reach a statistically-significant level to explain siblings’ profile membership. 

Sibling resemblance in profile membership was estimated with intraclass correlations, and values were moderate (0.46 ≤ *ρ* ≤ 0.55). Further, with increases in model complexity, i.e., from M_1_ to M_3_, resemblance tended to increase somewhat.

## 5. Discussion

To the best of our knowledge, this is the first study that aimed to identify children and adolescents´ multivariate profiles by focusing simultaneously on a set of health-related markers including lifestyle behaviors, physical fitness, and body composition using latent profile analysis. Furthermore, we have also investigated the putative links between these profiles with biological, sociodemographic, and built environmental characteristics along with the estimation of sibling resemblance in such profiles. Two distinct multivariate profiles emerged from our analysis. The differences between these two profiles were related to body composition, physical fitness, and unhealthy diet, which, in turn, can be interrelated with the development of non-communicable diseases during children’s and adolescents’ period of physical growth and development [1]. Additionally, these health-related markers apparently did not manifest in the same way and intensity in all subjects. For example, although 8 subjects (data not shown) from our sample shared the same physical activity score (0.26), they also differed substantially in other health-related markers, namely: unhealthy diet (−1.20–1.00), healthy diet (−1.00–2.02), physical fitness (−0.73–0.80), and body composition (−1.14–2.23). 

Previous research within the person-centered paradigm reported that different sets of health-related markers tended to cluster differently in subgroups of subjects who shared identical characteristics within the same group [36,41,42]. However, these reports were based on cluster or latent class analyses and were most often divergent. For example, Hartz et al. [43] using physical activity, sedentary time, and diet identified three distinct clusters for boys and girls aged 12–19 years, whereas Cabanas-Sanchez et al. [36] found five clusters based on moderate-to-vigorous physical activity, screen time, non-screen sedentary time, diet, and sleep. Further, Patnode et al. [14] studied three classes emerging from 12 behaviors related to physical activity, sports participation, and sedentary activities. Notwithstanding the relevance of previous studies, they apparently did not consider viewing youth from a broader perspective because their study relied mostly on the lifestyle behaviors, neglecting thus the significance of other important factors like physical fitness and body composition. 

Our data showed that profiles were mostly conditioned on the relationship between body fat, physical fitness, and unhealthy diet. Unexpectedly, however, youth with lower body fat and higher physical fitness also tended to show higher levels of unhealthy diet habits. This probably signposts that physical fitness exerts a protective effect on body fat increases while attenuating the effects of unhealthy diet. This was partially corroborated by the data of Garcia-Pastor et al. [44] showing that adolescents with higher body fat content also had lower levels of cardiorespiratory and muscular endurance fitness than their leaner counterparts, whereas the links between physical activity or diet with body fat was less evident. Furthermore, Artero et al. [45] reported that adolescents from nine European countries with better muscular fitness also had lower chronic inflammation most probably due to lower levels of fatness. However, overweight and obese adolescents displayed less adverse profiles if they maintained appropriate levels of muscular fitness. In sum, both physical fitness and body composition seem to be more important key markers distinguishing healthy from unhealthy profiles than physical activity, healthy diet, and lower levels of sedentariness. 

We also showed that, from sociodemographic characteristics, only father’s occupation and education were predictors of profile membership. Further, none of the potential built environment correlates, as well as biological maturation significantly predicted profile membership. Although there was evidence indicating that different facets of the built environment influenced physical activity, diet, body composition, and physical fitness [46,47,48,49], we were not able to identify a study that investigated these associations using multivariate health profiles. Furthermore, previous studies rarely explored, in the same article (and data analysis strategy), how likely different facets of the built environment are to influence health-related markers. As such, comparisons with our study are problematic. However, we were able to identify a study [50] that used latent class analysis to identify classes of neighborhoods based on built environment characteristics and then tested how adolescent physical activity, sedentary behavior, and screen time differed by neighborhood type/class [50]. The results were similar to those of our study, i.e., no differences in adolescent physical activity, sedentary behavior, and screen time by neighborhood class were observed. This calls for more research on multivariate health profile analysis to better understand the impact of the built environment on youngsters’ health [51]. 

Sociodemographic characteristics were previously associated with the co-occurrence of health behaviors [41,52,53]. For example, Fernandez-Alvira et al. [41] using children’s data from seven European countries found associations between parental education and cluster membership, i.e., children of lower educated parents were more likely to be allocated in the low activity/sedentary cluster than in the active cluster. Ottevaere et al. [52] also used cluster analysis and reported that adolescents from low educated parents had diets of lower quality and spent more time in sedentary activities. On the other hand, Liu et al. [53] examined adolescents’ co-varying patterns of physical activity and sedentary behavior and highlighted that adolescents pertaining to a class with moderate physical activity and high sedentary behavior were more likely to belong to low income families than their peers from other classes. In our study, we used a set of sociodemographic indicators, and our results are not without some controversy. Indeed, father’s occupation was positively associated with a better profile, but father’s education was negatively linked. This may suggest that treating both educational and occupational categories as indicators of the same dimension or fundamental cause might ignore their sometimes sizeable independent and distinct contributions to health. Furthermore, different socioeconomic indicators have different effects across life and obviously can influence health in different ways. For example, education level is typically addressed until early adulthood (approximately ages 18–25) with no further changes. Yet, final occupation level tends to be acquired later and is more prone to change than education level [54]. 

We identified a moderate sibling resemblance in their multivariate profile. In addition, with the inclusion of covariates, the intraclass correlations increased, suggesting a potential effect of the genetic and shared characteristics on these health-related markers. Regardless, we were only able to retrieve one study based on the person-centered approach that examined the co-occurrence of health-related markers in families [55]. Indeed, Niermann et al. [55] used cluster analysis to identify patterns of health behaviors within families based on triads (father, mother, and child) and reported lower intraclass correlation values than in our sample. Furthermore, previous studies investigating sibling resemblance for each individual health-related marker, namely physical fitness [17,56,57], physical activity [58,59,60], body composition [61,62], and diet [63], showed low-to-moderate sibling similarities. However, the effect sizes of such sib-ships’ likeness varied across studies, which can also be partially explained by differences in sample sizes and composition, age range, reference population, analytical strategies, and covariate adjustments. In spite of this, several genome-wide association (GWA) studies using health-related markers [64,65,66,67] reported their relationships with candidate genes. For example, Lightfoot [64] and de Vilhena e Santos et al. [65] performed systematic reviews and highlighted different genetic determinants involved in the regulation of physical activity, namely: dopamine receptor 1 (Drd1) and helix loop helix 2 (Nhlh2), as well as Ace, Gln223ARrg, MC4R, and DRD2, respectively. Further, a GWA approach also allowed Lu and Day [66] to identify 12 loci related to % body fat, whilst Willems and Wright [67] identified 16 loci associated with grip strength. Moreover, Schnurr et al. [68] found a common genetic etiology between whole % body fat and cardiorespiratory fitness. However, we could not find any genetic study related to the co-occurrence of all six health markers that could eventually lead us in to a deeper understanding of sibling resemblance in their profile membership. 

Notwithstanding the relevance of these results, our study has limitations: (1) participants were not recruited from all Portuguese regions, which limits the generalization of the results to the whole Portuguese population; yet, this situation is common in family and twin studies; (2) the use of questionnaires to obtain information about physical activity, screen time, healthy, and unhealthy diet is prone to errors, even though the questionnaires have been applied in controlled conditions. Further, these questionnaires are frequently used and have been shown to be reliable in previous studies [21,69,70,71]. 

## 6. Conclusions

In conclusion, two distinct multivariate profiles emerged from our analysis, and the differences between such profiles are related to body composition, physical fitness, and unhealthy diet. Moreover, none of the potential built environment correlates and biological maturation were associated with profile membership, but father’s occupation and education were. Finally, a moderate resemblance in sibling profiles was found and increased when the effects of sociodemographic characteristics and built environment were controlled for, which may mirror the effects of genetic make-up and shared characteristics. Taken together, these results point to the importance of targeting specific groups when developing adequate intervention programs to improve their health. Since the last century, many intervention programs were proposed and implemented to improve children’s and adolescents’ health. Yet, substantive frameworks underlying their elaboration and effectiveness are still unclear. Therefore, we claim that such intervention programs should be at least empirical evidence based, targeting different classes/profiles of adolescents. In other words, a risk-prevention program that is effective in one culture, population, or specific group may not be, or even much less so, effective in others.

## Figures and Tables

**Figure 1 ijerph-15-02799-f001:**
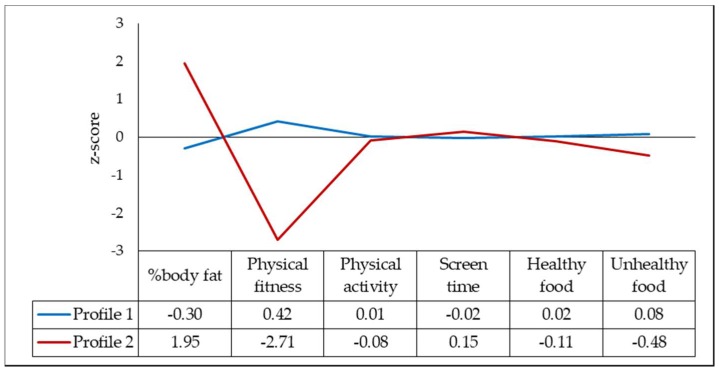
Profiles’ display for biological health markers, as well as lifestyle behaviors.

**Table 1 ijerph-15-02799-t001:** Sample descriptive characteristics (means, standard deviations (SD), and frequencies (%)) for sibling pairs.

	Brother-Brother (*n* = 200)	Sister-Sister (*n* = 167)	Brother-Sister (*n* = 369)	All Sibs
Variables	Mean ± SD	Mean ± SD	Mean ± SD	Mean ± SD
**Biological characteristics**				
Chronological age (years)	13.1 ± 1.8	12.6 ± 1.6	12.9 ± 1.7	12.9 ± 1.7
Biological Maturation (years)	−0.14 ± 1.9	−0.33 ± 1.3	−0.24 ± 1.6	−0.23 ± 1.6
**Body composition**				
Body fat (%)	20.0 ± 6.4	25.8 ± 5.6	23.2 ± 6.6	22.9 ± 6.7
**Physical fitness**				
Handgrip strength (kg^f^) *	27.0 ± 9.2	22.0 ± 5.5	23.9 ± 7.3	24.3 ± 7.7
Standing long jump (cm)	162.4 ± 31.6	133.7 ± 28.3	151.5 ± 31.1	150.4 ± 32.3
1-mile run/walk (min)	8.6 ± 2.0	10.2 ± 2.2	9.2 ± 2.2	9.3 ± 2.2
Shuttle-run (s)	11.2 ± 1.8	12.2 ± 2.1	11.4 ± 1.9	11.5 ± 1.9
**Lifestyle behaviors**				
Total physical activity	8.4 ± 1.5	7.7 ± 1.3	8.1 ± 1.5	8.1 ± 1.5
Screen time	3.6 ± 2.5	2.9 ± 2.1	3.4 ± 2.2	3.3 ± 2.3
Unhealthy diet	31.5 ± 16.0	23.9 ± 11.5	28.6 ± 14.6	28.3 ± 14.6
Healthy diet	57.6 ± 16.3	57.0 ± 15.0	57.8 ± 14.4	57.6 ± 15.1
**Sociodemographic characteristics**				
Occupation				
Mother	5.9 ± 2.4	6.1 ± 2.7	5.2 ± 2.7	5.6 ± 2.5
Father	5.9 ± 2.4	6.1 ± 2.4	5.4 ± 2.4	5.7 ± 2.6
Education	%	%	%	
Mother				
<Grade 12	28.8	48.0	31.7	34.7
Grade 12/diploma/technical qualification	42.9	24.0	36.0	39.1
University level	28.2	28.0	32.3	23.7
Father				
<Grade 12	32.0	48.6	34.8	37.1
Grade 12/diploma/technical qualification	36.0	40.3	40.3	39.1
University level	32.0	11.1	24.9	23.7
**Built Environment**				
Access to shops/markets				
Disagree	21.5	19.7	23.5	22.2
Agree	78.5	80.3	76.5	77.8
Access to public transportation				
Disagree	14.1	17.1	20.2	18.0
Agree	85.9	82.9	79.8	82.0
Traffic				
Disagree	54.8	70.4	62.8	62.4
Agree	45.2	29.6	37.2	37.6
Safety				
Disagree	59.3	65.8	58.3	60.4
Agree	40.7	34.2	41.7	39.6
Presence of sidewalk and bike paths				
Disagree	49.7	45.4	50.0	48.8
Agree	50.3	54.6	50.0	51.2
Recreational facilities				
Disagree	54.2	63.2	52.7	55.4
Agree	45.8	36.8	47.3	44.6

* kg^f^: kilogram-force.

**Table 2 ijerph-15-02799-t002:** Criteria used to identify the best number of latent profiles.

Fit Measures	Number of Profiles		
	1	2	3
No. of parameters	12	19	26
AIC	11,907.918	11,643.753	11,552.738
BIC	119,630.198	11,731.279	11,672.511
VLMR LRT	-	−5941.959	−5802.877
		1 profile vs. 2 profiles	2 profiles vs. 3 profiles
*p*-value	-	<0.001	0.217
Adjusted LRT test	-	272.277	102.792
*p*-value	-	<0.001	0.222

AIC: Akaike information criteria; BIC: Bayesian information criteria; VLMR: Vuong–Lo–Mendell–Rubin likelihood ratio test; LRT: Likelihood-ratio test.

**Table 3 ijerph-15-02799-t003:** Multilevel logistic regression in siblings profile membership.

	Model 1 (M_1_)		Model 2 (M_2_)		Model 3 (M_3_)	
Variables	Odds Ratio (SE)	95% CI	Odds Ratio (SE)	95% CI	Odds Ratio (SE)	95% CI
*Fixed effects*						
Intercept (P2) ^§^	0.07 (0.02) ***	0.04–0.13	0.01(0.01) ***	0.00–0.09	0.01 (0.01) ***	0.00–0.09
Maturity offset	1.18 (0.10) *	1.00–1.40	1.21 (0.13) ^ns^	0.99–1.48	1.22 (0.13) ^ns^	0.98–1.50
Father occupation			1.24 (0.12) *	1.02–1.51	1.24 (0.13) *	1.01–1.53
Mother occupation			0.93 (0.08) ^ns^	0.79–1.11	0.93 (0.09) ^ns^	0.78–1.12
Father education (Grade 12) ^∞^			3.18 (1.72) *	1.10–9.19	3.38 (1.94) *	1.10–10.41
Father education (university level)			6.40 (5.09) *	1.35–30.38	7.25 (6.13) *	1.38–38.02
Mother education (Grade 12) ^∞^			0.91 (0.48) ^ns^	0.33–2.55	0.92 (0.51) ^ns^	0.31–2.72
Mother education (university level)			0.79 (0.58) ^ns^	0.18–3.34	0.70 (0.55) ^ns^	0.15–3.29
Shops/markers (agree) ^¥^					1.70 (0.55) ^ns^	0.66–4.34
Public transportation (agree) ^¥^					0.86 (0.43) ^ns^	0.33–2.30
Traffic (agree) ^¥^					0.55 (0.22) ^ns^	0.26–1.19
Sidewalks/bike paths (agree) ^¥^					1.02 (0.36) ^ns^	0.51–2.05
Safety (agree) ^¥^					1.35 (0.48) ^ns^	0.67–2.73
Recreational facilities (agree) ^¥^					1.01 (0.38) ^ns^	0.48–2.11
*Random effects (variance components)*	Estimate (SE)		Estimate (SE)		Estimate (SE)	
Between siblings’	2.84 (1.17)		3.58 (1.51)		4.08 (1.76)	
	*ρ* (95% CI)		*ρ* (95% CI)		*ρ* (95% CI)	
Intraclass correlation	0.46 (0.28–0.66)		0.52 (0.32–0.71)		0.55 (0.35–0.74)	
Deviance	555.76		471.50		467.22	

* *p* < 0.05; ** *p* < 0.01; *** *p* < 0.001; ^ns^ = non-significant; ^§^ P1 is the reference; ^∞^ <grade 12 is the reference; ^¥^ not agree is the reference.

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
