# Peer review of "Profile Resemblance in Health-Related Markers: The Portuguese Sibling Study on Growth, Fitness, Lifestyle, and Health"

_ijerph, 2018, doi:10.3390/ijerph15122799_

Round 1

Reviewer 1 Report

The abstarct should begin with a brief introduction. The instruments used must be indicated. The keywords must be MeSH or DeCS. To affirm that there are no studies that relate on the relationship between lifestyle behaviors and training and body composition, it is necessary to explain with greater precision the literature review carried out, in what databases and what terms were searched for in them. The calculation of the sample should be clarified or at least indicate the universe with respect to the total collected. Taking into account that a regression analysis is included, as well as the calculation of risk, case and control groups are needed, which have not been clearly specified in the sample. In the methodology line 104 and 105 it is indicated that no oversignification was found, an aspect that should not be included in this section but in the results. It is of great interest to include the main validation data of the scales and instruments used. ORs have been included that have not been clearly explained in the statistical analysis of the methodology section. The conclusions should not include bibliographic citations. Review references, when cited as Vancouver, the titles of the journals should not appear in italics.

Author Response

We thank the reviewer for her/his comments and suggestions that will certainly improve the new draft. All changes in the new draft are marked in yellow.

The abstract should begin with a brief introduction.

Authors´ answer: The following was added:

The co-occurrence of health-related markers and their associations with individual, family and environmental characteristics have not yet been widely explored in siblings.

The instruments used must be indicated.

Authors’ answer: The following was added: Body fat was measured with a portable bioelectrical impedance scale; biological maturation was assessed with the maturity offset; handgrip strength, standing long jump, 1-mile run and shuttle run were used to mark physical fitness. Health behaviors, sociodemographic and built environmental characteristics were obtained by questionnaire.

The keywords must be MeSH or DeCS.

Authors’ answer: changed according to MeSH

Keywords: siblings; health behaviors; physical fitness; body composition; adolescents

To affirm that there are no studies that relate on the relationship between lifestyle behaviors and training and body composition, it is necessary to explain with greater precision the literature review carried out, in what databases and what terms were searched for in them.

Authors’ Answer:  The literature review was conducted in July 2018 within four bibliographical online databases, namely: PubMed, Sport Discus, Scopus and PsycINFO. The search terms consisted of two classes: health-related markers (body composition, physical fitness, physical activity, diet and screen time) and analysis (profile latent analysis, latent class analysis and cluster analysis). Regarding the search operation, we have combined the aforementioned search terms thus resulting in different search phrases. 

We think the review will not disagree with us if we do not include this information in the introduction as this is not a usual practice, unless a systematic review is undertaken which is not our goal.

The calculation of the sample should be clarified or at least indicate the universe with respect to the total collected.

Authors’ Answer: The sample used in the present paper is part of the Portuguese Healthy Family Study (Santos, et al. 2014). Further, since in this paper we focus on related individuals (siblings) the sampling procedure is quite different from others using unrelated individuals.  For example, when using twins usually the data comes from registries. Also, when dealing with families, they are selected based on some criteria and in no special way represent all the families within a region or a country. Yet, by no means have their results lacked some generalization if one bears in mind the specificities of the methodology. Therefore, the sampling strategy was to invite all siblings studying in the same schools (~4100 siblings) and the response rate was 80%. A total of 3825 siblings was previously indicated in the methodology section. 

In any case we added more information to clarify the sampling strategy:

The subjects in this study were part of a larger study, the Portuguese Healthy Family Study [17], in which participants were randomly sampled from schools in mainland (north and central) Portugal and from the Azores islands.  Those who had their siblings studying in the same school were invited to take part in the current study (~4100), with a response rate of ~80%. Written informed consent was obtained from legal guardians, and the project was approved by the Ethics Committee of the University of Porto and school authorities. The final sample consists of 3285 siblings 9-20 years of age. However, for the present study the sample comprises 736 biological siblings (350 females and 386 males) from 370 nuclear families with complete data on lifestyle behaviors, physical fitness and body composition.  No statistically significant mean differences were observed between included and excluded siblings in height, weight and % body fat.

References:

Santos DM, Katzmarzyk PT, Diego VP, Blangero J, Souza MC, Freitas DL, et al. Genotype by sex and genotype by age interactions with sedentary behavior: the Portuguese Healthy Family Study. PloS one. 2014;9(10):e110025

Taking into account that a regression analysis is included, as well as the calculation of risk, case and control groups are needed, which have not been clearly specified in the sample.

Authors’ Answer: Please note that this is not a case-control study, and no relative risk was calculated. Ours is a family study only involving siblings, with the intent of describing similarities and/or differences between siblings in terms of continuous outcomes.  Thus, we did not address questions of risk and/or case/control status.    

In the methodology line 104 and 105 it is indicated that no oversignification was found, an aspect that should not be included in this section but in the results.

Authors’ Answer:   We feel that it is important to report these non-significant results here because they relate to the composition of the sample used in the subsequent analyses of this paper, versus subjects that were not included in the analyses reported in this paper.   Our intent is to reassure readers that the included subjects are not statistically different than those not included in terms of these important variables, and thus to address this important issue regarding the sample used in this paper. 

It is of great interest to include the main validation data of the scales and instruments used.

Authors Answer: Additional information was added.

ORs have been included that have not been clearly explained in the statistical analysis of the methodology section.

Authors’ Answer: We have added the following sentence to clarify this aspect:

Using the best solution on profiles and given the nested structure of the data (subjects nested within sib-ships), we used a multilevel logistic model to predict profile membership (P1 is the reference) and sequentially tested three models of increasing complexity. Model 1 (M1) only included biological maturation; in model 2 (M2) we added father and mother education as well as their occupation; finally, model 3 (M3) included built environment covariates.

The conclusions should not include bibliographic citations.

Authors’ Answer: Corrected.

Review references, when cited as Vancouver, the titles of the journals should not appear in italics.

Authors’ Answer: Please note that the journal adopted the ACS style and the title of the journal should be written in italics.

“Journal Articles:
1. Author 1, A.B.; Author 2, C.D. Title of the article. Abbreviated Journal Name Year, Volume, page range.”

Reviewer 2 Report

The manuscript entitled "Profile resemblance in health-related markers. The Portuguese sibling study on growth, fitness, lifestyle and health" is a well-written document and entails a significant advance in the comprehension of the health-related behaviors in this population and their biopsychosocial determinants. Some minor issues should be addressed by the authors:

Abstract:

A little information about the background of the study is needed at the beginning of the Abstract. It is important to contextualize the reader before the introduction of the aims of the study.

Introduction

I recommend authors to classify better the information included in the introduction section. It is important to identify separately information about the lifestyles and about the environment. Probably the inclusion of subheadings could improve this issue.

I do not find necessary the inclusion of information of latent class analysis in this section. This information could be removed. 

Hypotheses of each described aim should be included at the end of the introduction section.

Results

In Table 1, the column All sibs should be included first, and then, the rest of the columns.

Discussion

Attending the contradictory results, authors should discuss deeply how individuals with lower body fat and more physical fit, exhibit higher levels of unhealthy diet.

In prediction analyses, environment seems not to be a significant predictor. Taking into account that authors includes the evaluation of these contextual variables as an advantage of the present study, this non-significant results should be discussed.

Author Response

The manuscript entitled "Profile resemblance in health-related markers. The Portuguese sibling study on growth, fitness, lifestyle and health" is a well-written document and entails a significant advance in the comprehension of the health-related behaviors in this population and their biopsychosocial determinants. Some minor issues should be addressed by the authors:

We thank the reviewer for her/his compliments on our work as well as all comments and suggestions made that will certainly improve the quality of this new draft. All changes in the new draft are marked in yellow.

Abstract:

A little information about the background of the study is needed at the beginning of the Abstract. It is important to contextualize the reader before the introduction of the aims of the study.

Authors’ answer: A similar comment was made by the first reviewer and the following was added:

The co-occurrence of health-related makers and their associations with individual, family and environmental characteristics have not yet been widely explored in siblings.

Introduction

I recommend authors to classify better the information included in the introduction section. It is important to identify separately information about the lifestyles and about the environment. Probably the inclusion of subheadings could improve this issue.

Authors’ answer: We thank the reviewer for this comment. Please note, however, that we wrote the introduction with an integrative frame of mind. We think the reviewer will agree with us with our approach to the subject. We could add subheadings such as residence, familial environment, clustering/profiling (and they are included in the new draft), but this may probably fragment the introduction.

I do not find necessary the inclusion of information of latent class analysis in this section. This information could be removed. 

Authors’ answer: Done.

Hypotheses of each described aim should be included at the end of the introduction section.

Authors’ answer: We thank the reviewer for this suggestion, and the following was added:

Based on our aims the following questions were addressed: (1) Can we identify multivariate profiles of health-related markers in youth? If so, how many will emerge?; (2) How are biological, sociodemographic and built environment characteristics associated with profile membership, i.e., how strong are their effect sizes? (3) How sizeable is sibling resemblance in their multivariate profiles?

Results

In Table 1, the column All sibs should be included first, and then, the rest of the columns.

Authors’ answer: Done.

Discussion

Attending the contradictory results, authors should discuss deeply how individuals with lower body fat and more physical fit, exhibit higher levels of unhealthy diet.

Authors’ answer: Please see the section on page 10 of the discussion. The combination of lower body fat, higher fitness and unhealthy diet patterns can co-exist – physical fitness provides a powerful protective factor against high levels of body fat, even in the face of poor dietary patterns. Please note that we only examined dietary patterns and we did not include a marker of dietary intake (i.e. kcal/day) and body fat would be expected to be more related to total dietary intake rather than dietary patterns, which we have shown in a previous study that dietary patterns were not related to obesity in 9-11 year old children (1.  Katzmarzyk, P.T.; Barreira, T.V.; Broyles, S.T.; Champagne, C.M.; Chaput, J.P.; Fogelholm, M.; Hu, G.; Johnson, W.D.; Kuriyan, R.; Kurpad, A., et al. Relationship between lifestyle behaviors and obesity in children ages 9-11: Results from a 12-country study. Obesity 2015;23(8):1696-702.).

In prediction analyses, environment seems not to be a significant predictor. Taking into account that authors includes the evaluation of these contextual variables as an advantage of the present study, this non-significant results should be discussed

Authors’ answer: We thank the reviewer for this important call, and the following was added:

Although there is evidence indicating that different facets of the built environment influence physical activity, diet, body composition and physical fitness [46-49], we were not able to identify a study that investigated these associations using multivariate health profiles. Furthermore, previous studies rarely explored, in the same article (and data analysis strategy), how likely different facets of the built environment are to influence health-related markers. As such, comparisons with our study are problematic. However, we were able to identify a study [50] that used latent class analysis to identify classes of neighborhoods based on built environment characteristics and then test how adolescent physical activity, sedentary behavior, and screen time differed by neighborhood type/class [50]. The results are similar to those of our study, i.e., no differences in adolescent physical activity, sedentary behavior and screen time by neighborhood class was observed. This calls for more research on multivariate health profile analysis to better understand the impact of the built environment on youngsters health [51]. 
